# Unravelling Factors Shaping International Students’ Learning and Mental Wellbeing During the COVID-19 Pandemic: An Integrative Review

**DOI:** 10.3390/ijerph22010037

**Published:** 2024-12-30

**Authors:** Huaqiong Zhou, Fatch Kalembo, Ambili Nair, Eric Lim, Xiang-Yu Hou, Linda Ng

**Affiliations:** 1Curtin School of Nursing, Curtin University, Perth 6102, Australia; h.zhou@curtin.edu.au (H.Z.); fatchwelcom.kalembo@curtin.edu.au (F.K.); ambili.nair@curtin.edu.au (A.N.); eric.lim@curtin.edu.au (E.L.); 2Broken Hill University Department of Rural Health, The University of Sydney, Sydney 2880, Australia; xiang-yu.hou@sydney.edu.au; 3School of Nursing & Midwifery, University of Southern Queensland, Ipswich 4350, Australia

**Keywords:** COVID-19 pandemic, international students, learning experiences, mental health, wellbeing

## Abstract

The international tertiary education sector was significantly affected by the COVID-19 pandemic due to the risk of negative learning and psychosocial experiences. Most international students who remained in the host countries demonstrated admirable resilience and adaptability during those challenging times. An integrative review of factors shaping international students’ learning and mental wellbeing during the COVID-19 pandemic was conducted. Five electronic databases—CINAHL, MEDLINE, ProQuest, PsycINFO, and Web of Science—were searched from 2020 to 2023 using the key search terms ‘international students’, ‘tertiary education’, ‘mental health and wellbeing’, and ‘COVID’. A total of 38 studies were included in this review. They revealed six factors across learning and psychosocial experiences. Predisposing factors for maladjustments included the students being younger and possessing poor English proficiency. Precipitating factors were related to online teaching/learning, and lack of accessibility and or insufficient learning and living resources. Perpetuating factors pertained to living arrangements. The protective factor identified was institutional support. This review highlighted that multifaceted factors were associated with international students’ experiences and mental health and wellbeing. In-depth understanding of risk and protective factors can help policymakers to prepare for unprecedented challenges and reduce disruptions to international students’ education and mental health when studying abroad.

## 1. Introduction

The COVID-19 public health pandemic impacted all sectors of the community, including universities [1]. As the global crisis escalated, international tertiary education was among the first global sectors that were significantly impacted, as many international students returned to their countries of origin [2]. The unavoidable social isolation and close-down policies not only caused abrupt changes in the landscape of teaching delivery and learning experiences but also impacted the psycho-social wellbeing of domestic and international students [3]. In particular, the three most reported mental health issues by students were anxiety [4,5], depression [6,7], and stress [8,9]. Up to 65.5% of students felt dissatisfied with their learning experience during the pandemic [10,11,12]. However, despite knowing that there would be curfews, border closures, and travel restrictions [13], a significant number of international students still chose to remain in their host countries to continue with their studies in undergraduate or postgraduate courses.

Even prior to the pandemic, international students were considered a vulnerable group as they were at risk of developing poor mental health due to language barriers, lack of support from families and friends, and the need to acculturate to unfamiliar educational and living conditions [14,15,16]. During the pandemic, these vulnerabilities were exacerbated as most international students who chose to remain in their host countries could only contact their families and friends by phone or online due to travel restrictions and lockdowns [14,17]. Their experience with long periods of physical distancing from their loved ones induced feelings of loneliness and homesickness [17]. Some students with Asian backgrounds also experienced racial discrimination in the host community, which caused them to doubt and regret their decision to remain in their host countries [14,18].

The pandemic also caused marked disruptions to normal learning and teaching patterns. For example, all or some of the face-to-face classes were transformed into online classes; clinical placements/field work were cancelled; examination and assessment formats were changed without the students being given much notice [1,2]. These rapid changes added to the burdens faced by international students, who had to develop digital literacy quickly, ensure internet access, and learn how to navigate online learning platforms while dealing with pre-existing vulnerabilities [1,2].

Despite having a higher risk of experiencing poorer learning, mental health, and wellbeing, most international students who remained in their host countries demonstrated admirable resilience and adaptability during these times [19]. However, resilience does not negate the complexities of their experiences as these students continued to navigate numerous academic, social, and emotional challenges. To date, no study has systematically reviewed the research evidence on factors affecting international students’ learning experiences and mental wellbeing. As such, the aim of this paper was to conduct a comprehensive literature review to yield an in-depth understanding of risk and protective factors for international students.

## 2. Materials and Methods

An integrative review was conducted, guided by a 5-stage framework by Whittemore and Knafl (2005) [20]. An integrative review is a method of research that appraises, analyses, and integrates literature on a topic so that new frameworks and evaluations are generated. This methodology allows the inclusion of studies with diverse data collection methods, including quantitative, quantitative, or mixed methods [21]. The PRISMA statement was also used to structure the review, minimise analysis bias, and systematically present findings [22].

### 2.1. Literature Search Strategy

The purpose of the narrative integrative review was to collate and synthesize research evidence from 1st January 2020 to 31st December 2023 that focused on international tertiary students’ (i) learning experiences and (ii) mental health and wellbeing during the COVID-19 pandemic. A search was conducted in the following five electronic databases: CINAHL, MEDLINE, ProQuest, PsycINFO and Web of Science. Database-specific subject headings and relevant text words were used. Search strategies contained terms related to (international or foreign*) AND (student* or postgrad* or “post-grad*” or undergrad* or “under-grad*) AND (university* or college*) AND (‘learning’ or ‘coping’ or ‘wellbeing’ or “well being” or wellness or “mental* health*” or “mental hygiene” or stress*) AND (coronavir* OR “corona virus*” OR betacoronavir* OR COVID-19 OR “COVID-19” OR “COVID-2019” OR COVID2019 OR pandemic).

The titles and abstracts of the downloaded papers were independently screened by two reviewers (HZ and EL) to identify studies that were eligible for inclusion. Papers were eligible for inclusion if they were (a) published in English, (b) peer-reviewed, (c) had clear evidence of research methodology, and (d) relevant to the purpose of the narrative integrative review. Subsequently, the full texts of the eligible papers were retrieved and read to confirm the number of papers to be included. All divergent opinions were resolved by a third independent reviewer (EL) (Figure 1). A search of unpublished academic theses from a university library and a manual search of the reference lists were also carried out to locate any potential studies for inclusion.

### 2.2. Quality Appraisal of Included Studies

The quality of the included papers was appraised independently by two authors (HZ and EL). Three types of studies (cross-sectional surveys, qualitative interview studies, or mixed-methods studies) were independently assessed using the mixed methods appraisal tool (MMAT) [23] (Hong et al., 2018). The MMAT systematically appraises the quality of studies by assessing research design, data collection method, data analysis, and interpretation of results.

### 2.3. Data Analysis and Narrative Synthesis

The 4Ps model of case formulation [24] was used to systematically categorise the extracted information from the included papers related to international tertiary students’ learning experiences and mental health wellbeing while living in their host countries during the COVID-19 pandemic. Using the 4Ps model of case formulation, manual coding was completed inductively based on the meaning of the identified factors and organised into (i) predisposing, (ii) precipitating, (iii) perpetuating, and (iv) protective factors. This model was selected as it allowed the researchers to categorise the data and provide a structured understanding of the risk and protective factors that contribute to international students’ experience. All of the authors checked and validated that the extracted information was sorted into the right category, and this step ensured the trustworthiness of the process. Subsequently, information extracted from each category was narratively synthesised to facilitate writing up the findings. This provided a more holistic understanding of the factors that influenced international students who stayed in their host country during the COVID-19 pandemic.

## 3. Results

### 3.1. Literature Search Results

A search of five databases generated 3765 records. Duplicated records (n = 426) were first removed, followed by excluding 3298 irrelevant records based on the title and abstract. Of the remaining 41 records, four were further excluded, as three were conference abstracts and one was a non-English publication. Full texts of the remaining 37 records were retrieved. Unpublished academic theses were also searched. A manual search of the reference lists of the 37 remaining papers identified one additional paper for inclusion. A total of 38 papers were included in the narrative integrative review.

### 3.2. Outcome of Critical Appraisal of Included Studies

An assessment of the included studies is presented in Table 1. All 38 included papers provided sufficient information on the study population, with selection criteria, outcome measurements, data collection methods, and data analysis. No studies were further excluded based on their quality.

### 3.3. Characteristics of the Included Papers

All 38 papers included in this narrative integrative review involved international undergraduate or postgraduate students who remained in their host countries to continue with their studies during the COVID-19 pandemic. The sample size of the included papers ranged from 20 [6] to 7199 [45]. Of the 38 studies, 15 were conducted in China, seven in the United States of America (USA), two each in Australia and Poland, and one each in Germany, Indonesia, Italy, Jordan, Korea, The Netherlands, New Zealand, Russia, Taiwan, Turkey, United Kingdom, and Ukraine (Table 2). Twenty-eight of the 38 included studies were cross-sectional surveys; six were qualitative studies conducted via focus group interviews (n = 3), one-to-one semi-structured interviews (n = 2), and biographical method/writing an essay (n = 1); and four were mixed methods using survey and focus groups.

### 3.4. Factors Influencing International Students’ Learning Experiences During the COVID-19 Pandemic

A total of 20 factors associated with the learning experiences of international students were extracted from three of the included studies. Four of the 16 factors were categorised as predisposing factors and contributed to reasons that can negatively impact the learning experiences of international students (Refers to Table 3). These four factors were (1) being female in the host country [29], (2) having English as their second language [26], (3) being in the basic year/beginning of their course [12], and (4) pressure from family [26].

Ten of the studies included in this integrative review provided evidence of six precipitating factors that led to international students having poor learning experiences. They were (1) lack of practical classes or lack of clinical placements [12,46], (2) lack of resources and support for research, laboratory, and library services due to curfews [25,26,46], (3) unstable and slow internet connection [11,26,36,46], (4) encountered difficulty navigating online platforms with insufficient support for online teaching [10,11,26,36,50,53], (5) feared for their own and their families’ health [12,36,50], and (6) had a feeling of uncertainty about study progression and career opportunity [12,28].

The disruptions and changes to teaching and learning activities highlighted two perpetuating factors experienced by international students, resulting in poor learning experiences. These two factors were (1) distractions in the home environment [36,53] and (2) diminished interest in online learning due to the absence of interaction with peers and lecturers [36,50,53].

Seven of the 38 included studies identified eight protective factors deemed to be critical for international students to maintain positive learning experiences during the COVID-19 pandemic. These protective factors were subcategorised into personal factors and institutional factors. Personal factors included (1) exercising self-discipline to participate in online learning activities [12], (2) perceiving self-isolation as opportunities to engage in new hobbies or connect with family/friends virtually [12,26,28,53], and (3) having social contact and sufficient COVID-19 knowledge [29]. Institutional factors included (1) receiving communication from academic staff [26], (2) having well-designed and easily accessible learning materials [12,28,36], (3) having flexible learning times [11,53], (4) receiving well-structured assessments [11,12,53], and (5) receiving individualised pastoral care and learning support from the school/ university [12,26,36].

### 3.5. Factors Influencing Mental Health and Wellbeing of International Students During the COVID-19 Pandemic

A total of 29 factors associated with the mental health and wellbeing of international students were extracted and grouped under the 4Ps factors. Nine of the 38 included articles identified three predisposing factors, including (1) being younger [4,9,27,35], (2) being female [9,29,34,39,43,48,49], and (3) having English as a second language [26].

Twenty-one of the included articles reported nine factors that could have precipitated poorer mental health and wellbeing of international students who stayed in the host country during the pandemic. These factors were as follows: (1) had enrolled in undergraduate courses or non-health related majors [4,37,45], (2) severity of the impact of the pandemic on the host country [7,31,41], (3) being diagnosed with COVID-19 [5,30], (4) feared for personal and family’s health [9,32,33,34,38,41], (5) felt disconnected and lonely due to curfews and social isolation [6,27,30,34,35,39,48,51,52], (6) changed from in-class to purely online teaching [8,30,35,51], (7) had feelings of uncertainty of study progression and career trajectory [6,41,49,52], (8) experienced acculturating stress [33], and (9) experienced anti-Asian sentiments [6].

Nine of the included studies identified five perpetuating factors that were associated with the mental health and wellbeing of international students during the COVID-19 pandemic. The five factors were (1) living arrangements: either in the metropolitan region of the host country, in a shared house with other people, or living alone and in rural areas [4,5,8,34,41,52]; (2) financial hardships because they were unable to work due to curfews or lacked monetary support from family [5,9,30,34,45]; (3) excessive media exposure to COVID-19 information and government management plans [5,41]; (4) close-contact with people positive for COVID-19 [5]; and (5) negative coping styles [45,47].

Sixteen articles identified six institutional and six personal factors that were perceived by international students as protective of their mental health and wellbeing during the COVID-19 pandemic. The data analysis contributed evidence that international students experienced fewer psychological issues, such as anxiety and/or depression, when their enrolled institutions (1) responded appropriately to the pandemic [44], (2) provided them with flexible visa arrangements [33], (3) had a designated international student support team [8], (4) provided them with living/financial relief [6,33,48], (5) provided them with timely and reliable information [6,33,37,38], and (6) provided sufficient resources for teaching, such as a library or flexible learning and examination systems [6,8,33] and research equipment and laboratories [33].

Six personal protection factors associated with decreased risks of experiencing anxiety and/or depression included (1) receiving social support from peers and family via virtual communication platforms [8,30,31,33,40,41,48]; (2) living arrangements involving living with family [38,45] or in big cities [35]; (3) having tuition fees paid by family or borrowing from others [45]; (4) being positive [40,41,44,47] and engaging in exercise, worship, cooking, watching COVID-19 news, and playing video games [39]; (5) seeking professional advice when experiencing negative emotions [30,42,52]; and (6) enrolling in postgraduate courses or health-related majors [34,37].

## 4. Discussion

Thirty-eight studies were included in this review and revealed a total of 37 factors that affected the learning, mental health, and wellbeing of international students who remained in tertiary studies during the COVID-19 pandemic. The six consistently reported factors were identified and grouped based on the 4Ps model of case formulation across learning and psychosocial experiences. Predisposing factors for maladjustments included students being younger and possessing poor English proficiency. Precipitating factors were related to online teaching/learning and lack of accessibility to or insufficient learning and living resources. Perpetuating factors pertained to living arrangements, while protective factors included institutional support.

The findings of this integrative review highlighted multifaceted factors that affected the learning and mental wellbeing of international students during the COVID-19 pandemic. The predisposing and precipitating findings showed that moving online created a challenge, particularly with the language and the content, navigating the online medium, learning how to participate virtually, and the expectation to adapt and evolve with the changes [54]. The lack of staff consistency in online teaching [54] could be part of the reason why online teaching was reported to be detrimental to learning in different countries [10,11,26,36,50,53]. Contrary to the findings of our review, a Canadian study reported that international students enjoyed online learning compared to domestic students during the pandemic [55]. The reasons for this are unknown.

Precipitating the online connectivity challenges, our findings showed the capacity of students to maintain motivation to learn [12,36,50] and self-discipline [56]. One of the protective factors perceived by international students was that they felt supported by the lecturers and that the learning materials were well designed [12,28,36]. Chien-Yuan and Guo, 2021 [56] also reported that learner–content interaction had the most significant impact on students’ learning outcomes and satisfaction. Using breakout rooms gave the students an opportunity to interact and engage with other students, filling a void in social isolation experienced due to COVID-19 lockdown requirements [57,58]. Innovative interaction techniques were particularly evident in the simulation space when moving online. Clinical skills that involve auditory and visual training can be effectively imparted through digital tools, while those that depend on manual dexterity necessitate in-person practice for effective skill consolidation [59]. This aligned with our findings that healthcare students saw the value of face-to-face clinical placement in receiving high-value education and did not find simulated online learning to be beneficial [46]. There was a need for lecturers to develop innovative modes of teaching clinical skills online [57,60]. In medicine, virtual reality showed a simulation of real-life clinical experiences, and in midwifery, skills were demonstrated online, with the use of scripts as different partners in the midwife–woman dyad. Students played the roles in a group online session, ending in a plan of care being developed [57,60].

Not surprisingly, the learning challenges faced by international students were related to their mental health and wellbeing and were thematically similar. The factors are similar to the literature in countries such as the USA, Europe, and Asia. An Australian study described international students as being considered particularly vulnerable to experiencing poor mental health when studying away from their country of origin [61]. The paper suggested that this is reasonable, considering that students face new and varied academic and cultural challenges and that international university students had higher levels of depression compared to domestic students [62].

Our findings from the 38 papers found similar student experiences of mental well-being. Predominantly, the findings of this study show that students experienced loneliness and isolation from social contacts [6,30,35,39,48,51,52], although not specifically from family or friends. There were comments that isolation was linked to a loss of face-to-face class interaction [8,30,35,51], which is also supported by a Chinese study that found that resources should be appropriate for students’ varying levels of literacy [63]. Contrary to our expectations, an Australian study reported that domestic students were more lonely compared to international students during the COVID-19 pandemic [64]. This could be partly due to the findings that international students who received better social support during the COVID-19 pandemic adjusted better both academically and psychologically [48].

Of note, students were worried about their progression in the program of study, and this was reinforced by Hawley et al., 2021 [65], who highlighted that this was true for courses related to health sciences, where it was difficult to get practical skills while the teaching was online. This made the students feel inadequately trained and prepared in terms of sufficient employability skills for their discipline, which led to uncertainty about job prospects following the completion of their education, as companies had reduced the number of internships or new graduate positions [65].

Financial hardship is discussed in the literature as a barrier to mental wellness and includes literacy in navigating the health system [63]. This was not evident in our findings; however, we delved deeper to reveal that students were concerned with contracting COVID-19 from those who had tested positive [5,45]. Our data revealed that financial stress was associated with accommodation, whether it be living in a shared arrangement or alone, and that this was closely aligned with the inability to work due to the lockdown curfews or to get a reliable source of external revenue [5,9,30,45].

We found that many authors suggested varied resources, including libraries, flexible learning spaces, research facilities, and examination techniques [6,8,33]. This is similar to the opinion of Larcombe et al., 2023 [61] and Huang et al., 2020 [63], who claim institutions have a responsibility to provide learning support for this cohort and that this is a protective factor in preserving mental wellness. Huang et al., 2020 [63] also commented that universities could provide more cross-culture activities to enhance awareness of cultural diversity on campus. As students had no physical contact with family overseas and limited capability to communicate online [8,12,30,31,33,43,48], it is reasonable to suggest that institutions should provide reliable platforms to facilitate interaction with loved ones overseas.

The findings of this review have significance in practice, particularly given the ongoing challenges faced by international students nearly five years after the COVID-19 restrictions. While emergency measurements implemented during the COVID-19 pandemic have largely subsided, the persistence of hybrid learning models and the normalisation of online teaching highlights the continued need for support. For example, it is critical to provide support for international students, including easy access to and navigation of online teaching platforms or resources, such as workshops to orient students to the new online teaching technologies and resources; to establish and keep open lines of communication with students; to identify and develop student-friendly support resources for online teaching and learning technologies; to introduce student hubs with stocked computers on campus, especially when students have poor internet connectivity in their places of residence; to facilitate the establishment of student peer support systems for new and existing international students to reduce loneliness and isolation. Universities should establish student mental wellbeing clinics within campuses where students can access mental health services at no cost or at a small fee. Furthermore, international students should be supported in building resilience and equipped with the skills needed to manage future disruptions or uncertainties in their academic and personal lives. The findings of the review emphasise the importance of universities developing student-centred and inclusive support systems to effectively address sudden disruptions and ongoing challenges. Therefore, future research could focus on identifying and evaluating effective strategies that could help international students to maintain mental health and wellbeing.

## 5. Conclusions

This integrative review of research evidence offers a more holistic understanding of factors that influence international students’ learning experience and their mental health and wellbeing. It provides insightful evidence for policymakers to improve support for international students in learning and mental health during a pandemic such as COVID-19. Future research could be conducted with international students to achieve a better understanding of their coping strategies.

## Figures and Tables

**Figure 1 ijerph-22-00037-f001:**
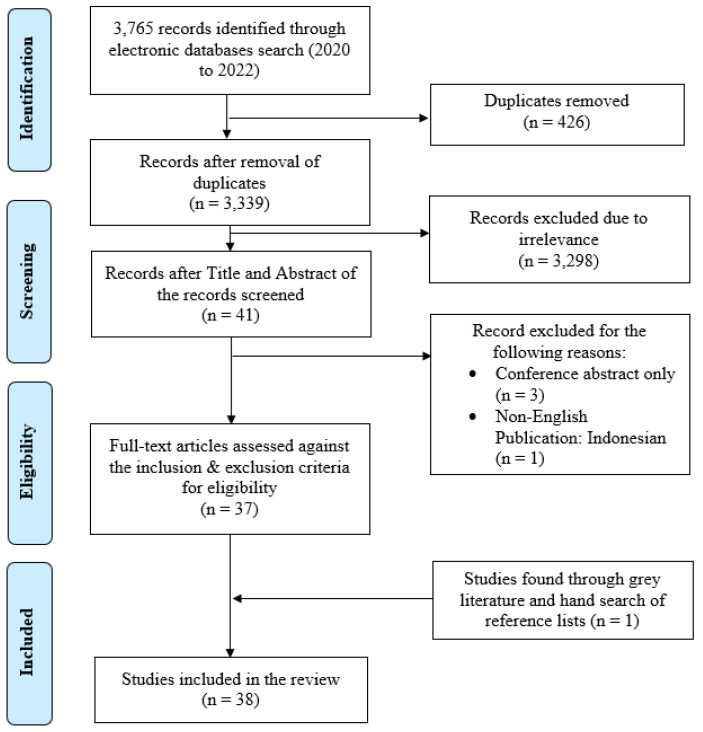
Flow chart of the search and study selection process (PRISMA).

**Table 1 ijerph-22-00037-t001:** Quality assessment of the 38 included studies using MMAT.

1st Author, Year Country	Qualitative					Qualitative Descriptive				
	1.1. Is the Qualitative Approach Appropriate to Answer the Research Question?	1.2. Are the Qualitative Data Collection Methods Adequate to Address the Research Question?	1.3. Are the Findings Adequately Derived from the Data?	1.4. Is the Interpretation of Results Sufficiently Substantiated by Data?	1.5. Is There Coherence Between Qualitative Data Sources, Collection, Analysis, and Interpretation?	4.1. Is the Sampling Strategy Relevant to Address the Research Question?	4.2. Is the Sample Representative of the Target Population?	4.3. Are the Measurements Appropriate?	4.4. Is the Risk of Nonresponse Bias low?	4.5. Is the Statistical Analysis Appropriate to Answer the Research Question?
Abdul-Rahaman, 2022 [25] Russia						▲	▲	▲	n/a	▲
Al-Oraibi, 2022 [26] UK	▲	▲	▲	▲	▲					
Brown, 2022 [10] Australia						▲	▲	▲	n/a	▲
Brownlow, 2022 [27] Australia	▲	▲	▲	▲	▲	▲	▲	▲	n/a	▲
Dong, 2022 [28] USA						▲	▲	▲	n/a	▲
Erden, 2022 [29] Turkey						▲	▲	▲	n/a	▲
Fu, 2022 [6] USA	▲	▲	▲	▲	▲					
Jagroop-Dearing, 2022 [8] New Zealand	▲	▲	▲	▲	▲	▲	▲	▲	n/a	▲
Jiang, 2022 [11] China	▲	▲	▲	▲	▲					
Lin, 2022 [30] USA						▲	▲	▲	n/a	▲
Lu, 2022 [31] China						▲	▲	▲	n/a	▲
Kivelä, 2022 [32] Netherlands						▲	▲	▲	n/a	▲
Mbous, 2022 [33] USA	▲	▲	▲	▲	▲					
Nadareishvili, 2022 [34] USA						▲	▲	▲	n/a	▲
Tan, 2022 [35] China						▲	▲	▲	n/a	▲
Tian, 2022 [36] China	▲	▲	▲	▲	▲	▲	▲	▲	n/a	▲
Tozini, 2022 [37] USA						▲	▲	▲	n/a	▲
Alam, 2021 [4] China						▲	▲	▲	n/a	▲
Ahorsu, 2021 [38] Taiwan						▲	▲	▲	n/a	▲
Almomani, 2021 [39] Jordan						▲	▲	▲	n/a	▲
Lai, 2021a [12] China	▲	▲	▲	▲	▲	▲	▲	▲	n/a	▲
Lai, 2021b [40] China	▲	▲	▲	▲	▲					
Lai, 2021b [41] China	▲	▲	▲	▲	▲					
Leshchyna, 2021 [42] Ukraine						▲	▲	▲	n/a	▲
Li, 2021a [12] China						▲	▲	▲	n/a	▲
Li, 2021b [43] China						▲	▲	▲	n/a	▲
Khan, 2021 [44] Italy						▲	▲	▲	n/a	▲
Kim2021 [5] Korea						▲	▲	▲	n/a	▲
Negash, 2021 [45] Germany						▲	▲	▲	n/a	▲
Shoukat, 2021 [7] Indonesia						▲	▲	▲	n/a	▲
Song, 2021 [9] China						▲	▲	▲	n/a	▲
Trzcionka, 2021 [46] Poland	▲	▲	▲	▲	▲					
Wang, 2021 [47] China						▲	▲	▲	n/a	▲
Wilczewski, 2021 [48] Poland						▲	▲	▲	n/a	▲
Yuan, 2021 [49] China						▲	▲	▲	n/a	▲
Chirikov, 2020 [50] USA						▲	▲	▲	n/a	▲
Lai, 2020 [51] China						▲	▲	▲	n/a	▲
Wang, 2020 [52] China						▲	▲	▲	n/a	▲
Zhang, 2020 [53] China						▲	▲	▲	n/a	▲

Black triangle ▲: The criterion was assessed as being met; n/a: Not applicable.

**Table 2 ijerph-22-00037-t002:** Characteristics of the 38 included papers.

1st Author, Year Country	Method of Data Collection	Number of Participants (N=) Characteristics of Participants	Outcome Measures
Abdul-Rahaman, 2022 [25] Russia	Survey	N = 394 International doctoral students in Russia.	Overall satisfaction with the general quality of learning by language of proficiency, aspects of university support, aspects of university support by language of proficiency during the COVID-19 crisis, and distance learning. Scale of 1 to 5 (1 = “absolutely dissatisfied” to 5 = “fully satisfied”).
Al-Oraibi, 2022 [26] UK	Focus groups	N = 46 8 groups of international students in England (n = 29) University staff supporting international students (n = 17)	Focus groups explored international students’ and university staff’s views of COVID-19 restrictions, self-isolation, and their wellbeing and support needs.
Brown, 2022 [10] Australia	Survey	N = 151 Occupational therapy students in Australia. Majority of domestic students were full-time females aged 20–24 years. 38 of 151 were international students from the Asia–Pacific region	Perceptions and experiences of online learning during the COVID-19 pandemic by online self-report questionnaire (demographic data, hours per week of direct face-to-face educational sessions, direct and indirect online sessions, and employment). Students’ engagement with and experiences of online learning: SELES and DELES standardized self-report instruments.
Brownlow, 2022 [27] Australia	Survey, Focus groups	N = 231 HDR students at a regional university. 47 (20.3%) international students Male/Female = 5.9:4.1	Brunel Mood Scale (BRUMS)—24 mood descriptors on 6 domains: Tension, Depression, Anger, Vigour, Fatigue, and Confusion. 11 participants in 3 focus groups (61–71 min) on experiences
Dong, 2022 [28] USA	Survey	N = 177 Asian international graduate students in the United States (aged 29.4 ± 5.11 years). 50.8% female (n = 90). Masters, Doctoral, and Postdoctoral exchange students.	Depressive symptoms: Patient Health Questionnaire-9 (PHQ-9). Generalised Anxiety Disorder Screener GAD-7 Scale. Social isolation: Loneliness: Social subscale of the Loneliness Scale. Distress about social distancing: 1 item of All of Us Research Pro ram COVID-19 Participant Experience (COPE) Survey “Since COVID-19 outbreak, have recommendations for socially distancing caused stress for you?” Rating “A lot, somewhat, a little or not at all”. Home boundness: 6 Items from the COPE Survey; A 4-point. Likert scale (None of the days, a few days, most days, every day). Everyday discrimination: 6 Items from the COPE Survey. Perceived racial discrimination: 1 item “What forms of racism incidents have you, your family, friends, or significant others experienced since the COVID-19 outbreak?”
Erden, 2022 [29] Turkey	Survey	N = 7363 from 9 universities 233 (3%) international students Male/Female = 7.1:2.8 UG 74%	The COVID-19 International Student Wellbeing Study (C19 ISWS) 7 modules: Sociodemographic information; Study-related information; Before and during COVID-19 outbreak; COVID-19 diagnosis, symptoms and perceived worries; Stressors, informal support, and mental wellbeing; Student-specific questions and concerns; and COVID-19 knowledge and information.
Fu, 2022 [6] USA	Phenomenology	N = 20 Chinese Doctoral students in the United States. Male/Female = 10:10.	In-depth interviews on students’ experiences in areas of international study, life, and career trajectory; Institutions’ response and support to Anti-Asian movements.
Jagroop-Dearing, 2022 [8] New Zealand	Survey Semi-structured interviews	N = 43 Postgraduate health and nursing international students in New Zealand (aged 31.3 ± 7.3 years). Male/Female = 12%:88%. From India (32, 74%), Philippines (6, 14%), Sri Lanka (3, 8%), Nepal (1, 2%), and Brazil 1, 2%). 29 (67%) worked as essential health workers in aged care facilities, hospitals, and pharmacies.	Survey Perceived level of stress: Adapted Questioner from Short Form of the Perceived Stress Scale (PSS-10). Semi-structured interview: Sources of perceived stress and coping measures -qual. Utilised general wellbeing. Challenges faced during the epidemic. How working as an essential worker during the epidemic affected the workers.
Jiang, 2022 [11] China	Semi-structured interviews	N = 40 Medical international students in China (second to sixth academic year). Male/Female = 18:22. From India (n = 16), Bangladesh (n = 7), Nigeria (n = 4), and Sri Lanka (n = 3).	Semi-structured interviews on factors influencing students’ academic success.
Lin, 2022 [30] USA	Survey	N = 1881 Chinese international students in the United States (age 21.39 ± 2.48 years). Male (n = 976, 51.9%). 97.8% were not in the USA when completing the survey. Undergraduates (n = 1302, 69.2%), Masters (n = 508, 27%), and PhD (n = 71, 3.8%).	Self-constructed single-item questions on workload, frequency of remote learning, and levels of agreement with the impacts of remote learning on personal relationships, academic performance, and future careers. Social Support Rating Scale (SSRS). Insomnia Severity Index (ISI). Depressive symptoms: Patient Health Questionnaire-9 (PHQ-9). Generalised Anxiety Disorder screener (GAD-7).
Lu, 2022 [31] China	Survey	N = 519 International medical students in China (50.4%, aged 22.76 ± 3.60 years). 451 (86%) not in China when completing survey. 453 (87.3%) undergraduates. 409 (78.8%) contracted COVID-19.	Depressive symptoms: PHQ-9. COVID-19-pandemic-related stress: 5 questions on the 4-point Likert scale (Perception of COVID-19 outbreak, worry about being infected, worry about family/relatives/friends being infected, worry about exam scores, worry about not being able to complete the study). Coping: Simplified Coping Style Questionnaire (SCSQ). Perceived social support: Perceived Social Support Scale (PSSS) questionnaire.
Kivelä, 2022 [32] Netherlands	Survey	N = 349 Two cohorts of international students in the Netherlands: March 2020 (n = 207). March 2021 (n = 142). Age: Year 2020 (aged 20.3 ± 2.19 years). Year 2021 (aged 21.1 ± 5.61 years). Undergraduate students: Year 2020 (n = 202, 98%). Year 2021 (n = 131, 92%). International students: Year 2020 (n = 81, 39%). Year 2021 (n = 88, 62%).	Depressive symptoms: Beck Depression Inventory (BDI-I). Suicidal ideation: Beck Scale for Suicidal Ideation (BSSI). Anxiety symptoms: Beck Anxiety Index (BAI). Post-traumatic stress disorder (PTSD) symptoms: PTSA Checklist for DSM-V (PCL-5). Alcohol (ab)use: Alcohol Use Disorders Inventory test (AUDIT). Insomnia symptoms: Insomnia Severity Index (ISI). Academic stress: Law Student Perceived Stress Scale (LSPSS). Loneliness: De Jong-Gierveld Loneliness Scale (DJGLS). Fear of COVID-19: Fear of COVID-19 Scale (FVC-19S). Coping with COVID-19: Pandemic Coping Scale (PCS).
Mbous, 2022 [33] USA	Focus groups	N = 13 International students in the United States who did not have parents/guardians living (aged 26.3 ± 3.8 years). Male/Female = 4:8 (one missing data). Undergraduate students (n = 11, 84.6%).	1.5–2 h per focus group interview followed a standard script to document challenges international students encountered while studying during the pandemic.
Nadareishvili, 2022 [34] USA	Survey	N = 984 All universities in Georgia 224 (22.8%) international students Medical students (46.6%) Male/Female = 7.1:2.8	STAI: Anxiety CES-D: Depression RASS: Suicidality
Tan, 2022 [35] China	Survey	N = 1090 International students in China (aged < 22 = 529, aged > 22 = 561). Male/Female = 255:835. Single = 756.	Anxiety-related psychological problems: GAD-7 score.
Tian, 2022 [36] China	Survey Semi-structured interviews	N = 1010 Undergraduate and postgraduate international students in China. Male/Female = 598:412.	Interviews: Thematic analysis on perceptions of the emergency online learning, online learning environment, and online learning engagement. Survey: A 5-point Likert scale for student engagement factors and online learning environment factors on online learning satisfaction.
Tozini, 2022 [37] USA	Survey	N = 359 International students of 66 different nationalities in the United States. Male/Female/Others = 36.8%:62.1%:1.2%. 40% Bachelor’s, 26% Master’s, 32% Doctoral degree, and 7% other programs. China (23.7%), India (10.3%), Mexico (5.6%), South Korea (3.6%), and Saudi Arabia (3.1%).	Social support, student-life stress, financial wellbeing, and satisfaction with the university’s handling of the COVID-19 crisis measured on a 5-point Likert scale.
Alam, 2021 [4] China	Survey	N = 402 International students in China. Male (n = 340, 84.6%). 18–40 years old (32.1% 18–25 years).	Symptoms of depression, anxiety, stress: The English version of the Depression Anxiety Stress Scale (DASS-21). Insomnia: Insomnia Severity Index (ISI). Psychological distress: Kessler Psychological Distress Scale (K6). Loneliness: University of California, Los Angeles, Loneliness Scale (UCLA-LS). Fear: Fear of COVID-19 (FCV-19S) scales.
Ahorsu, 2021 [38] Taiwan	Survey	N = 529 International students in Taiwan. Aged ≥ 20 years old and above.	Online survey on perceived susceptibility, sufficiency of resources, source of pandemic information, and support satisfaction. State–rait Anxiety Inventory tool. Suicidal ideation: 1 item “In the past seven days, did you have a suicidal idea”.
Almomani, 2021 [39] Jordan	Survey	N = 585 Local students and international students in Jordan. Female = 372 (63.6%). 171 (29.2%) were international students. 553 (94.5%) were undergraduate students. 228 (38%) were away from families. 434 (74.2%) studied medical and pharmaceutical degrees, 96 (16.4%) studied general sciences, 47 (8%) studied engineering, and 7 (1.2%) studied literacy and humanities.	Impact of curfew restrictions on mental health, including psychological symptoms and coping strategies. Common psychological symptoms: anxiety, sleep problems, depression, short temper, low self-esteem, feeling helpless, feeling worthless, feeling unhappy, and focusing problems.
Lai, 2021a [12] China	Survey, Focus group interviews	N = 91 Hong Kong students in the United Kingdom. Focus group interviews: 16 students. 38.5% male, 93.4% aged 18–25 years. 87.9% were undergraduates, 35.2% were final-year students, and 58.2% were in a medical/health-related course.	Psychological impact of COVID-19 survey. Perceived public attitudes and personal belief and practice in relation to facemask wearing: 3-Question Survey. Stress in relation to facemask wearing and facemask-related stressors 2- Question survey.
Lai, 2021b [41] China	Focus group interviews	N = 20 Full-time Chinese international students from Hong Kong in the United Kingdom or United States. Aged 18–25. 65% were undergraduates.	Semi-structured interview (1.5 h) on students’ stressors, cognitive appraisals, coping, and outcomes based on Transactional Model of Stress and Coping Theory.
Leshchyna, 2021 [42] Ukraine	Survey	N = 103 International students in Ukraine (aged 22.0 ± 1.8 years). Male/Female = 52:51. 3 groups of students: third year (n = 23), fourth year (n = 40), and fifth year (n = 40).	Psychoemotional state: PHQ-9 screening test. Scale of nervous and mental tension. The Hamilton Anxiety Rating Scales. The Hamilton Depression Rating Scales.
Li, 2021a [12] China	Survey	N = 1045 International Chinese college students in the United States. Male/Female = 535:475. Aged 18–23.	Substance abuse, self-injury, and suicidal thoughts.
Li, 2021b [43] China	Survey	N = 230 Medical and nursing international students in China. Students: Male/Female = 82:148. Studied MBBS (n = 207), Nursing (n = 23). Asian countries (n = 130, 56.5%), African countries (n = 100, 43.5%). Inside China (n = 77, 33.5%), outside China (n = 13, 66.5%).	Satisfaction with online education during the COVID-19 pandemic.
Khan, 2021 [44] Italy	Survey	N = 180 International students in China. Male/Female = 66.11%:33.89%. Married/unmarried = 23.89%:76.11%. Aged 20–29 (67.78%), over the age of 29 (32.22%). Studied PhD (50%), Masters (36.7%), and undergraduate (13.3%). Asian origin (68.3%), African (17.2%), and others (14.25%).	Impact of trust in university management on students’ psychological distress and their self-quarantine behaviour. Trust in university’s management team: 10-item 5-point scale (2 items/each subdimension: benevolence, integrity, competence, identification, and concern. Anxiety: A 5-point Likert scale and the Clinical Anxiety Scale, which consisted of 10 items. Acceptance of self-quarantine using a 5-point scale.
Kim2021 [5] Korea	Survey	N = 488 International students in South Korea. Male/Female = 41.8%:58.2%. Aged 25–30 years (63.9%). Undergraduate students (32.6%) and graduate students (67.4%). Living with someone (27.3%). Monthly income of ≤ USD 883 (72.3%). Resided in an urban area (23.2%) or a rural area (76.8%). Good vs. poor level of understanding of the host country’s media (56.6% vs. 43.4%). Used the media to obtain COVID-19-related information for at least two hours per day (29.5%).	Anxiety symptoms: Generalised Anxiety Disorder Scale (GAD-7). Depression symptoms: Patient Health Questionnaire (PHQ-9).
Negash, 2021 [45] Germany	Survey	N = 7199 International students in Germany. Age (24 years ± 4.7). Male/Female/Others = 30%:69%:1%. Relationship status: single (n = 3040), in a relationship (n = 3870), complicated (n = 289). Living in accommodation with others (n = 3255), alone (n = 1099). Studied Bachelor’s (n = 3485), Master’s (n = 1527), or Doctoral degrees (n = 327). Tuition fees paid by scholarship (n = 1093), others (n = 2665), with a loan (n = 197), and a combination of the above (n = 975).	Change in Financial Situation During COVID-19: 5-point Likert scale: Sufficient financial resources before and during the COVID-19 pandemic to cover monthly costs. Depressive Symptoms: Epidemiological Studies Depression Scale (CES-D 8 scale). Factors associated with worsened financial situation and/or depressive symptoms—questionnaire on 1. Sociodemographic; 2. Study-related factors, e.g., field of study; 3. Academic frustrations, e.g., increased workload, poor quality of education; 4. Social interaction; and 5. Finances.
Shoukat, 2021 [7] Indonesia	Survey	N = 86 International Pakistani students in Indonesia. Aged < 25 (n = 44), 25–30 years (n = 32), >30 years (n = 10). Male/Female = 59:27. Self-rated financial status: High (n = 14), Good (n = 16), Medium (n = 20), and Weak (n = 36).	Google forms via email and social media groups. Consisted of (i) closed and open-ended questions that examine repeated homesickness experiences and the severity of homesickness, (ii) scales that assess lethargy, fear of contracting coronavirus, social media pressures with a lot of hoaxes, cognition–emotion and academic motivation, and (iii) symptoms of anxiety such as fatigue, irritability, trouble sleeping or staying awake, panic attacks, excessive worrying, and restlessness.
Song, 2021 [9] China	Survey	N = 261 International Chinese students in the United States. Male/Female = 46.7%:53.3%. Aged ≤ 19 (50.6%), 20–23 (33.7%), >23 (15.7%). Came from China (47.8%), and living in the United States (52.1%). Self-reported health: Healthy (46.3%), average (39.4%), poor/very poor (14.1%). Self-reported economic pressure: None/little (45.5%), Moderate (36%), Severe (18.3%). Not sure about academic plan (50.5%), continue studying abroad (36.3%), going back to China (13%).	Psychological impact: post-traumatic stress disorder (PTSD) and Checklist Civilian Version (PCL-C). Mental health status: Depression, Anxiety, and Stress Scale (DASS).
Trzcionka, 2021 [46] Poland	Biographical method/essays	N = 40 Fifth-year dentistry students in Poland: international (n = 20), local (n = 20).	Possible losses and benefits subjectively felt by individuals due to the COVID-19 pandemic, adaptation to online learning, students’ opinions regarding change of habits, and psychological impact of the lockdown. Students write essays covering topics on a given framework.
Wang, 2021 [47] China	Survey	N = 453 International students in China (aged 22.09 ± 2.73 years). Male/Female = 233:220.	Psychological functioning, cross-cultural adaptation, confidence in COVID-19 control, Perceived Stress Scale, Resilience Scale, and the Life Orientation Test.
Wilczewski, 2021 [48] Poland	Survey	N = 357 International students from 62 countries in Poland. Stayed in the host country: N = 236. Male/Female = 77:58. Studied a Bachelor of Science (n = 133), Master of Science (n = 85), PhD (n = 16), and Other (n = 6). Stayed in home country: N = 121. Male/Female = 35:86. Bachelor of Science (n = 87), Master of Science (n = 29), PhD (n = 3), Other (n = 3).	Levels of loneliness: De Jong-Gierveld Loneliness Scale (DJGLS). Life and academic satisfaction: Satisfaction with Life Scale (SWLS). Acculturative stress: Acculturative Stress Scale. The online learning experience/academic adjustment, performance, loyalty: A self-developed 12-item scale measuring 4 aspects of online learning.
Yuan, 2021 [49] China	Survey	N = 519 International medical students in China (50.4%, aged 22.76 ± 3.60 years). 451 (86%) not in China when completing survey. Were undergraduates (n = 453, 87.3%). Contracted COVID-19 (n = 409, 78.8%).	Anxiety: Generalized Anxiety Disorder Assessment (GAD-7). Depressive symptoms: Patient Health Questionnaire-9 (PHQ-9). Coping: Simplified Coping Style Questionnaire (SCSQ). Stress: Perceived Stress Scale (PSS-10). Perceived social support: Multidimensional Scale of Perceived Social Support (MSPSS). Optimism: Revised Life Orientation Test (LOT-R). Resilience: Resilience Scale-14 (RS-14).
Chirikov, 2020 [50] USA	Survey	N = 3405 International students in the United States. Undergraduates (n = 1982, 8.8%), graduates/professionals (n = 1423, 18.5%).	Student Experience in the Research University (SERU) Consortium.
Lai, 2020 [51] China	Survey	N = 124 Living in the UK or US. Aged 18–25. Male: 36.3%.	Perceived stress, anxiety, insomnia, family functioning, resilience, and stress-coping strategies of international students studying abroad.
Wang, 2020 [52] China	Survey	N = 153 International students in China. Male/Female = 97:46. Were undergraduates (n = 19), Masters/postgraduate (n = 91), and PhD (n = 33) students.	Depression: Patient Health Questionnaire-9 (PHQ-9). Anxiety, quarantine, sleep duration, degree of worry about graduation, and sense of security: Generalized Anxiety Disorder-7 (GAD-7) scale + State–Trait Anxiety Inventory (STAI).
Zhang, 2020 [53] China	Survey	N = 84 International students enrolled in Medicine UG, Bachelor of Surgery (MBBS) Program in China. Completed pre-test survey (n = 50) or post-test survey (n = 48). Only 48 were included in the analysis.	Perception of online Traditional Chinese Medicine (TCM) course. Attitude, cognition, and behaviour towards TCM. An electronic self-administered questionnaire with 28 items related to the experience of online delivery of TCM. Items in the questionnaire used a 5-point Likert scale.

**Table 3 ijerph-22-00037-t003:** Summary of main findings of the 38 included papers.

(Year)	Learning Experiences and Mental Health and Wellbeing	4Ps Categories: Predisposing, Precipitating, Perpetuating, Protective
Abdul-Rahaman, 2022 [25] Russia	Dissatisfaction with the quality of learning.	**Precipitating:** Lack of research support services. Lack of laboratory equipment. Lack of access to required software. Lack of library resources.
Al-Oraibi, 2022 [26] UK	Fear that the qualifications would not be valued in their home country if conducted fully online. Increased anxiety related to future education and career prospects. Loneliness, anxiety, worry, sadness, and low mood during self-isolation. Disconnected from the university due to little or no contact from university administrators, personal tutors, or lecturers and lack of social connection.	**Predisposing:** English as a second language. Perceived pressure from family to do well in university. **Precipitating:** Unfamiliar with using online platforms. Online-based learning was not as effective as face-to-face lectures. Uncertainties about assignments or fieldwork, including laboratory sessions or placements. Perceived lack of support in self-isolation. Poor internet connectivity. Difficulties getting essential supplies. Lack of clear communication from the university on COVID-19-related changes. **Protective:** Ability to view self-isolation in a more positive light, as it allowed time to engage in new hobbies and talk to family and friends by phone or online. Pastoral care provided by university staff. Engaged in activities such as watching movies, chatting with family members and friends, reading, doing home exercises, and learning new hobbies. Ability to view online activities facilitates networking among students. Efforts from academic staff to make regular contact with students. Efforts from academic staff to individualise attention and personalise support.
Brown, 2022 [10] Australia	International students rated low scores in: Levels of self-directed learning in terms of decision-making, time management, and active learning capabilities. Significantly greater difficulties in the application of knowledge beyond the university setting. Enjoyment. Interactions with instructors. Psychological motivation.	**Precipitating:** Online learning. Limited learning opportunities. Postponement or cancellation of placements, fewer placement hours. Changes to community activities resulting from the switch to online platforms.
Brownlow, 2022 [27] Australia	48.16 for Vigor to 53.12 for Tension: Referred to as elevated risk of mental health issues.	**Predisposing:** Younger age International students vs. local students **Precipitating:** Balancing life demands and studying Fear of writing Isolation from peers
Dong, 2022 [28] USA	Depression symptoms, of which 20% had higher levels of depression and 12% had higher generalised anxiety.	**Precipitating:** Experienced perceived racial discrimination: racial insults, deliberate avoidance, getting spat at/sneezed or coughed on, and/or physical intimidation. Social distancing and loneliness. **Protective:** Homeboundness decreased depression symptoms.
Erden, 2022 [29] Turkey	Academic satisfaction. Wellbeing. COVID-19 knowledge.	**Predisposing:** Females **Protective:** Social contact COVID-19 knowledge
Fu, 2022 [6] USA	Helpless and loss of control about future possibilities of legally staying in the U.S. Depression.	**Precipitating:** Quarantine/social isolation. Disconnected from other students, advisors and the real world. Homesick and missing family members. Anti-Asian wave increased participants’ emotional stress and fear of being physically hurt. Delay research progress from discontinued fieldwork or wet experiments in laboratories. **Protective:** Supportive and understanding supervisors. Provision of a flexible examination system. Provision of flexible visa arrangements. Provision of a pandemic relief fund at the beginning of the epidemic. Provision of regular and consistent information.
Jagroop-Dearing, 2022 [8] New Zealand	International students reported stress associated with fear of contracting the COVID-19 virus.	**Precipitating:** Lack of face-to-face teaching and adapting to online learning. **Perpetuating:** Separation from family. **Protective:** Being an essential worker provided income and opportunities for social interaction. Availability of social media to communicate with family members and peers. Provision of institutional support via video conferencing calls, library learning services, international student support teams, and international student mentor programme.
Jiang, 2022 [11] China	The quality of online teaching and learning was compromised. Most international students expressed emotional fatigue, boredom, demotivation, and laziness when faced with online learning.	**Precipitating:** Problems with online learning during the pandemic. Recorded lectures rather than livestream lecturing. Had the benefit of meeting teachers face to face for Q and A sessions. Internet disconnection or slow speeds and technical problems Hindered online learning and taking online exams. **Protective:** Flexible learning time and access to videos.
Lin, 2022 [30] USA	24.5% of the international students had a PHQ-9 score ≥ 10. 20.7% of the international students had a GAD-7 score ≥ 10.	**Precipitating:** A higher risk of depression was related to the following: Previous traumatic events. Agreed with the pandemic’s negative impacts on financial status. Agreed with the pandemic’s negative impacts of remote learning on personal relationships. Higher Insomnia Severity Index (ISI) score. A higher risk of anxiety was related to the following: Previous traumatic events. High workloads include staying up for online classes. Agreed with the negative impacts of remote learning on personal relationships. Higher ISI score. **Protective:** Lower risk of depression was related to the following: Disagreed with the negative impacts of remote learning on academic performance and future careers. Keen to seek professional help with emotional issues. Higher Social Support Rating Scale (SSRS) score. Lower risk of anxiety was related to the following: Keen to seek professional help with emotional issues. Higher SSRS score.
Lu, 2022 [31] China	Mild to severe depressive symptoms.	**Precipitating:** COVID-19-pandemic-related stress was positively associated with depressive symptoms. **Protective:** Lowered PHQ-9 score when the students: Lived with family or friends. Stayed in cities without a COVID-19 outbreak. Perceived social support was negatively associated with depressive symptoms.
Kivelä, 2022 [32] Netherlands	Academic stress depressive symptoms, suicidal ideation, anxiety, PTSD, and loneliness.	**Precipitating:** Increased fear of COVID-19. Poor coping with COVID-19.
Mbous, 2022 [33] USA	Loneliness; no motivation, focus, and productivity; mental health issues; worry; fear; stress; anxiety; negative emotions; relationship dynamics; Sense of belonging; and uncertainty.	**Precipitating:** Residency challenges: Acculturation, bureaucracy, and administrative hurdles; discrimination; visa policies; travel restrictions; job search; and political climate. Lifestyle changes: Work environment, pandemic-induced attitudes, life balance. Negative effects: Concerns for relatives, homesickness. **Protective:** Social support: Family, friends, and university. University structure: Action, directives, and structural guidance. Centralisation of authority at the department/advisory level. Financial support. Flexibility or rigidity of programs. Graduation and research delays. Output requirement. Paradigm for research and teaching. Relationship with faculty.
Nadareishvili, 2022 [34] USA	Anxiety, depression, and suicidality	**Predisposing:** Females **Precipitating:** Poor general health condition Decrease physical activity Had a history of self-harm Family conflicts Fear of COVID-19 **Perpetuating:** Finances (anxiety and depression) **Protective:** Lower substance use International students coped better. Medical students (lower risk of depression).
Tan, 2022 [35] China	Anxiety	**Predisposing:** <22 years old. **Precipitating:** Purely online teaching. Socialised < 3 times/month. **Perpetuating:** Living in big cities.
Tian, 2022 [36] China	Perceptions of emergency online learning on value and dissatisfaction.	**Precipitating:** Qualification: Postgraduate international students were more satisfied with online learning satisfaction than undergraduate students. Major: Students enrolled in arts, humanities, and social sciences reported significantly higher levels of satisfaction than those in life sciences and medicine. Poor internet connectivity. Slow internet speed. Time difference between host and home countries for those who returned home. Distractions at home. Online teaching, thus, no interaction with teachers and peers. Activities reduced due to pandemic. Teachers insufficiently prepared for online teaching. **Perpetuating:** Lack of focus due to online teaching and being at home Non-conducive online learning environment due to being at home Lack of interest and boredom due to no interaction **Protective:** Well-designed education courses. A student-centred approach is adopted in the delivery of courses.
Tozini, 2022 [37] USA	**Undergraduate**: Lower levels of social support than for international doctoral students. More dissatisfied with the way their institution responded to the COVID-19 crisis than international postgraduate students. **Masters level**: A greater disruption in their ability to achieve academic goals than either UG or PhD.	**Precipitating:** Pandemic-related campus closures impacted students’ perceptions of social support and satisfaction with their university’s handling of the crisis Qualifications: Undergraduate studies **Protective:** Undertaking postgraduate studies Satisfaction with the universities’ responses Financial wellbeing was maintained by having multiple sources of funding for studies.
Alam, 2021 [4] China	Depression, anxiety, stress, insomnia, psychological distress, loneliness, and fear.	**Predisposing:** Aged 26–30 years. **Precipitating:** Enrolled in Engineering, Business, Social Sciences, Law, and/or Language courses. **Perpetuating:** Lived with roommates. Stayed in the host country < 2 years. Lived in the central region of the host country.
Ahorsu, 2021 [38] Taiwan	Anxiety. Suicidal ideation.	**Precipitating:** Perceived susceptibility to illness. **Protective:** Living with family members. Perceived satisfaction with support. Information seeking—perceived sufficiency with resources.
Almomani, 2021 [39] Jordan	Unhappiness, distress-like symptoms; for example, low self-esteem, hopelessness, helplessness, worthlessness, and depression. Anxiety, sleep problems, and short temper.	**Predisposing:** Females were more likely to experience psychological symptoms than males. **Predisposing:** 18–25 years/young students with unhappiness and distress-like symptoms. Older students (>26 years) with anxiety, sleep problems, and short temper. **Precipitating:** Negative impact of curfew restrictions on mental health of female and younger students. **Protective: Stress management by** Females: sleeping, studying, and worshipping. Males: working, exercising, and playing video games. Female and younger students: focusing on studying, exercising, worshipping, cooking, and watching related news.
Lai, 2021a [12] China	Stress.	**Protective:** Peer and family support: important for facing difficulties Positive thinking and adaptability: effective stress management.
Lai, 2021b [41] China	Negative impacts: feeling worried and stressed in the early stage of the pandemic. Positive gains: personal growth and enhanced family communication and relationships.	**Precipitating:** Severity of the outbreak. Worries about personal health. Difficulty locating personal protective equipment, COVID-19 prevention and management. Unclear arrangements from universities. **Perpetuating:** Lack of social support (such as unavailability of air tickets to fly back home). Uncertainties about their academic programme, unclear COVID-19-related information and ineffective local outbreak management were the most commonly reported. **Protective:** Emotion regulation: Talking with family and friends to ease loneliness. Connecting with friends going through the same situation to feel more understood. Practising positive thinking. Problem management: Students took proactive steps in using alternative measures to protect themselves when meeting challenges during the pandemic. Family, peers, and schools were the three main pillars of support for students and helped provide psychological support and relief from the challenges brought on by the pandemic.
Leshchyna, 2021 [42] Ukraine	High level of maladaptation.	**Protective:** Professional advice. Mandatory psychological assistance and a readaptation program.
Li, 2021a [12] China	Most students were dissatisfied with their online education experiences.	**Predisposing:** Living in host country Basic years of study **Precipitating:** Lack of practical classes (for those students who were in the clinical years). Uncertainty of returning to university (opening). Financial constraints/ economic issues related to the pandemic. Severity of the COVID-19 situation. Distance. Continent of origin and location of current residence **Protective:** Well-designed assignments. Internet speed/connectivity to access the internet for online learning. Support and help from the school during the online learning process. Self-discipline. Good course resources.
Li, 2021b [43] China	Suicide attempts, self-injury ideation, moderate depression, and minor anxiety levels.	**Predisposing:** Male students with anxiety problems were more likely to use medicine and less likely to take alcohol. Female students with anxiety problems were more likely to take medicine and drugs and smoke cigarettes but were less likely to take alcohol.
Khan, 2021 [44] Italy	Anxiety text score = 2.76 ± 0.790. Knowledge of COVID-19 score = 2.928 ± 1.104.	**Protective:** Those with knowledge of COVID-19 disease were more likely to accept self-quarantine. Practise prevention measures. Higher levels of trust in university management. Higher acceptance of self-quarantine.
Kim2021 [5] Korea	Sleep problems. Mild to moderate level of anxiety symptoms. Mild to severe depression symptoms.	**Predisposing:** Young age. A lower education level. Male sex. **Precipitating:** Good level of understanding of media. Rural residence. Experienced COVID-19 symptoms. **Perpetuating:** Living with someone. Clinical experience from a confirmed case of COVID-19. Income levels. Longer times of media usage on COVID-19.
Negash, 2021 [45] Germany	A worsened financial situation during the pandemic.	**Precipitating:** Migrates. Parents not being academics. Financial resources to cover monthly costs. Self-funded student (payment of tuition fee). Living alone. Studying a non-health-related field. **Perpetuating:** Worsened financial situation: Losing part-time jobs. Unpaid leave or ↓ working hours allowance. **Protective:** Tuition fee paid by others. Single, living with parents, or sharing accommodation. Studied health-related courses. Doctoral/PhD students Engaged in state exam programs such as medicine. Can borrow money from at least one person.
Shoukat, 2021 [7] Indonesia	Mental health crisis. Homesickness, anxiety, and depression.	**Predisposing:** Females—More homesickness. Males—More anxious. <25 years highly susceptible to anxiety and homesickness. >30 years were less vulnerable to anxiety and depression. **Precipitating:** Severity of COVID-19 pandemic.
Song, 2021 [9] China	Depression. Anxiety.	**Predisposing:** Aged 20–23 years (higher score). Female (lower score). **Precipitating:** Staying in host country. Poor health status. Severe economic pressure.
Trzcionka, 2021 [46] Poland	Need for a quick adaptation to a variety of e-learning platforms and systems, self-motivation for online courses or accessibility of the internet.	**Predisposing:** Final-year dentistry student who required practical placements. **Precipitating:** Lack of practical classes, limited to online theoretical classes. Social distancing (no contact with peers). Fear for own, relatives’, and friends’ health. Closure of borders—unable to help families if they contracted coronavirus. Decreased number of classes and in-person consultations (final-year students). Quality of the Internet connection.
Wang, 2021 [47] China	Anxiety and depressive symptoms.	**Predisposing:** Gender: Female. **Precipitating:** Stressors in school. **Perpetuating:** Negative coping style. Perceived stress. Staying up late. Current place of residence—host country.
Wilczewski, 2021 [48] Poland	Stress.	**Perpetuating:** Cross-cultural adaptation was negatively associated with the Global Severity Index (GSI) of Symptom Checklist 90 (mental health symptoms severity). **Protective:** Optimism and confidence in COVID-19 control.
Yuan, 2021 [49] China	Social and emotional loneliness.	**Predisposing:** Predominantly female (ratio 2:1). Older students tended to stay in the host country. **Precipitating:** Self-isolation. Lack of social support. Lack of communication with peers. **Protective:** Students in their home country. Fostering peer communication in online learning through group work and collaborative learning. Develop compensatory support services.
Chirikov, 2020 [50] USA	Healthcare and immigration issues during the pandemic were the greatest concerns for international students. Concerned about having financial support and obtaining health services. 25% of international students were concerned with xenophobia, harassment, or discrimination.	**Precipitating:** Lack of motivation for remote learning. Lack of interaction with other students. Inability to learn effectively in an online format. **Perpetuating:** Students from China generally adapted well or very well to remote instructions compared to students from India or Mexico.
Lai, 2020 [51] China	Perceived stress, severity of symptoms of anxiety and depression, and severity of symptoms of insomnia associated with each other.	**Predisposing:** Females. **Precipitating:** Personal health. Lack of social support. Females had lower resilience. Changes in teaching/learning format. Health of family and friends.
Wang, 2020 [52] China	Depression and anxiety. Worried about graduation. Feeling insecure.	**Predisposing:** Imposing confinement. Impact of the virus on their research. Future employment. Worry about graduation/ academic delays. **Protective:** High-quality internet. Strengthening psychological counselling and emotional remedy for graduating students.
Zhang, 2020 [53] China	Online learning combined with face-to-face learning in the future. More than half of international students preferred attending face-to-face classes.	**Perpetuating:** Perceived as less engaged by instructors. Easy distraction with the online format. The absence of hands-on activities. Students lack self-discipline. **Protective:** Ability to control the instructional pace of online lectures. Complete course events asynchronously at their convenience. Good quality instructional material and resources, technical support, accessible learning contents, and chapter highlights and summaries.

## Data Availability

The original contributions presented in this study are included in the article. Further inquiries can be directed to the corresponding author(s).

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
