# Peer review of "Unravelling Factors Shaping International Students’ Learning and Mental Wellbeing During the COVID-19 Pandemic: An Integrative Review"

_ijerph, 2024, doi:10.3390/ijerph22010037_

Round 1

Reviewer 1 Report

Comments and Suggestions for Authors

Thank you for your great work on systematic analysis of papers about student's well-being during COVID. I guess all representatives of academia evidenced difficulties and advantages that had students both who decided to return to home countries and who decided to stay inthe host countries.

There are some minor comments that could probably give some accents to your paper:

1. You said that you screen first titles and abstracts. Did you analyze full papers after that? Did you have any difficulties reaching all the papers? Did you consider papers in language different from English?

2. I thonk you could give more discussion about the effects fo COVID on experiences of students. While you consider COVID just as a solial isolation, it also affected health and emotional state of everyone, including students. Did any of the papers discuss that appraisals that students gave could be affected to the uncertainly and general fear fo what was going on? And the situation could be quite different in different countries in terms of limitations? For example, students from China who remained in host countries had more opportunities as restrictions in most countries were not that hard and that long as they were in China.

Reviewer 2 Report

Comments and Suggestions for Authors

Thank you for the opportunity to review the article, Unravelling factors shaping international students' learning and mental wellbeing during the COVID-19 pandemic: An integrative review. This paper presented a well conducted literature review of international students experiences during the COVID-19 pandemic when residing at their Universities during the lock down period. 

Introduction

-              Now that the pandemic has been declared “over” in many countries, should this section be written in the past tense? It is confusing in the tense it is presently written in as it sounds at times that we are still in a lockdown. 

-              Page 2 line 50, remove the from “up to 65.5% of the students” 

-              Context for which we are talking about international students would be helpful, undergraduates, graduate students, both? 

-              The introduction could benefit from some further synthesis, it felt disjointed at times. 

Materials and Methods

-              Please add the citation for the PRISMA statement

-              Page 3 line 105 should be were retrieved instead of was retrieved

-              What is “grey literature”?

-              Throughout the article there is inconsistent spacing between he (n=) and (n = )

-              Tables 2 and 3 are hard to read with the bullet  point formatting, there is too much space between the bullet and the text. Fonts are inconsistent in the tables as well

-              The protective factors would be helpful to be presented in some type of figure if possible

Discussion

-              When there is a list of three items, the third should have a comma after it (see page 16 line 276; “USA, Europe, and Asia). Please fix errors throughout

-              COVID—19 has an extra dash (page 16 line 291)

-              What are the implications of these data now, situated almost 5 years post the start of the lock down? What does this mean in the context of COVID-19 on students? 

Acknowledgements section is blank

The author Ambili Nairand name is spelled differently in the CREDiT statement.
